# Genetic Susceptibility to Hepatocellular Carcinoma in Patients with Chronic Hepatitis Virus Infection

**DOI:** 10.3390/v15020559

**Published:** 2023-02-17

**Authors:** Tsai-Hsuan Yang, Chi Chan, Po-Jiun Yang, Yu-Han Huang, Mei-Hsuan Lee

**Affiliations:** Institute of Clinical Medicine, National Yang Ming Chiao Tung University, 155 Li-Nong Street, Section Peitou, Taipei 112, Taiwan

**Keywords:** liver cancer, hepatitis B, hepatitis C, genome-wide association study, host

## Abstract

Hepatocellular carcinoma (HCC) is one of the leading causes of cancer-related deaths globally. The risk factors for HCC include chronic hepatitis B and C virus infections, excessive alcohol consumption, obesity, metabolic disease, and aflatoxin exposure. In addition to these viral and environmental risk factors, individual genetic predisposition is a major determinant of HCC risk. Familial clustering of HCC has been observed, and a hereditary factor likely contributes to the risk of HCC development. The familial aggregation may depend on a shared environment and genetic background as well as the interactions of environmental and genetic factors. Genome-wide association studies (GWASs) are one of the most practical tools for mapping the patterns of inheritance for the most common form of genomic variation, single nucleotide polymorphisms. This approach is practical for investigating genetic variants across the human genome, which is affected by thousands of common genetic variants that do not follow Mendelian inheritance. This review article summarizes the academic knowledge of GWAS-identified genetic loci and their association with HCC. We summarize the GWASs in accordance with various chronic hepatitis virus infection statuses. This genetic profiling could be used to identify candidate biomarkers to refine HCC screening and management by enabling individual risk-based personalization and stratification. A more comprehensive understanding of the genetic mechanisms underlying individual predisposition to HCC may lead to improvements in the prevention and early diagnosis of HCC and the development of effective treatment strategies.

## 1. Epidemiology of Hepatocellular Carcinoma

Liver cancer ranks sixth as the most commonly diagnosed cancer and the third leading cause of cancer-related death [1]. Among all cancers, liver cancer accounts for 4.7% of new cancer cases and 8.3% of cancer deaths. Liver cancer incidence has stabilized in men after decades of a steep increase. However, it continues to rise in women by more than 2% annually [2]. Liver cancer caused approximately 30,000 deaths in 2021 [2] and resulted in the loss of 12.5 million disability-adjusted life-years after diagnosis [3]; these statistics indicate its significant burden.

Hepatocellular carcinoma (HCC) is the most common histological type of liver cancer, accounting for over 75% of all primary liver cancers globally [4]. In general, HCC incidence and mortality rates are roughly equivalent because of the disease’s poor prognosis. The substantial variation in the incidence and mortality rates of HCC across geographical regions is attributed to differences in environmental exposure levels and timing, chronic hepatitis virus infection prevalence, access to healthcare resources, and the ability to early detect and promptly treat cancer [5]. Most HCC cases are estimated to occur in low-income or middle-income countries, particularly in Eastern Asia, Southeast Asia, and Sub-Saharan Africa.

The majority of HCC cases occur in patients with underlying liver disease, most of which result from chronic hepatitis B virus (HBV) and hepatitis C virus (HCV) infection or excessive alcohol consumption. Universal HBV vaccination programs and the implementation of direct-acting antiviral agents against HCV may change the etiological landscape of HCC. The prevalence of these two major risk factors of HCC may decline in the coming years. The increasing prevalence of nonalcoholic fatty liver disease (NAFLD), which, together with obesity and metabolic syndrome, elevates the risk of developing HCC and acts as a critical contributor to HCC after HBV and HCV infections have been controlled [6,7]. Estimates show that the increase in the average body mass index from 1991 to 2011 resulted in the incidence of liver cancer more than doubling.

In addition to viral and environmental risk factors, individual genetic predisposition may affect the risk of developing HCC. The influence of genetic factors was supported by considerations of heritability [8,9] and familial aggregations. Multiple genetic studies, using either the candidate gene approach or the genome-wide approach, have identified potential variants associated with HCC. In this article, we review the evidence from epidemiological studies as well as the genetic risk factors for HCC. As the genome-wide association study (GWAS) is a practical method of investigating genetic variants across the human genome, which is affected by thousands of common genetic variants that do not follow Mendelian inheritance, we summarize the findings of relevant GWASs, focusing on the impact of chronic hepatitis virus infection status on genetic loci associated with HCC.

## 2. Risk Factors for HCC

### 2.1. HBV

Chronic HBV infection is particularly prevalent in the Asia-Pacific region and in Sub-Saharan Africa, where the infection is predominantly acquired during the perinatal period or in early childhood. HBV has been classified as a human carcinogen by the International Agency for Research on Cancer [10]. Approximately 5% of the world’s population, or between 240 million and 350 million people, have chronic HBV infection. HBV has a primary role in the etiology of HCC. The incidence of HCC among carriers seropositive for hepatitis B surface antigen (HBsAg) is much higher compared to those seronegative for HBsAg (1158 vs. 5 per 100,000 person-years), resulting in a relative risk (RR) of 223 [11]. In addition, cirrhosis and HCC accounted for 54.3% and 1.5% of all deaths among HBV carriers and noncarriers, respectively [11]. Among individuals with chronic HBV, hepatitis B e antigen (HBeAg) [12] and the serum HBV DNA level are associated with a long-term risk of HCC development [13]. These seromarker measurements are essential for diagnosing, treating, and monitoring patients. Moreover, the HBV genotype C and specific variants of the basal core promoter and precore are associated with HCC development independent of the serum HBV DNA level [14]. Quantitative HBsAg level is also associated with HCC in a dose-response manner, particularly in patients seropositive for HBsAg with a low HBV DNA level (<10^6^ copies/mL) [15]. Several new markers have been identified as being associated with HCC risk after follow up [16,17], providing insights into the treatment of patients through antiviral therapy.

### 2.2. HCV

HCV is widely acknowledged as a major cause of chronic liver disease and the demand for liver transplantation. Patients with chronic HCV infection are often asymptomatic, which can result in unawareness of their illness until it progresses to end-stage liver disease. HCV infection plays a role in tumorigenesis through repetitive damage, regeneration, and fibrosis, and approximately 90% of HCV-associated HCC cases are preceded by liver cirrhosis [18]. Liver cirrhosis occurs in 20% to 30% of patients with chronic HCV infection after two to three decades [19]. Once cirrhosis has developed, approximately 1–4% of these patients will develop HCC each year [20]. The detectable serum HCV RNA level is a marker indicating the active replication of HCV, and 65% to 80% of serum samples seropositive for anti-HCV have detectable serum levels of HCV RNA [21,22]. Long-term follow-up studies have indicated that patients with elevated serum HCV RNA levels have a higher risk of HCC and liver-related death compared to those with undetectable levels, reinforcing the importance of antiviral treatment for those with chronic HCV infection [23,24]. In addition to RNA levels, HCV genotype one is prevalent in most geographical regions associated with an increased risk of HCC [25]. Seromarkers, including glycoprotein [26], immunological proteins [27], and circulating bile acids [28] are also associated with HCC risk, and they can assist in the early detection of HCC and subsequent intensive care. The treatment options for HCV infection have undergone significant improvements with the introduction of direct-acting antivirals. These drugs offer a new avenue for HCV treatment, inspiring hope that patients with chronic HCV infection can be cured in an effective and straightforward manner. Although treatment-induced RNA clearance is associated with a considerable reduction of the risk of HCC, the absolute risk of HCC remains high, which implies that ongoing surveillance for HCC may be necessary [29].

### 2.3. Alcohol Consumption

Excessive alcohol consumption is another risk factor for HCC [30]. A meta-analysis revealed that people who consume considerable amounts of alcohol have an approximately twofold greater risk of developing liver cancer [31] and that the relationship may vary by sex. Alcohol is associated with a nearly fourfold increased risk in women compared with a 59% increased risk in men [31]. Alcohol consumption mediated the association between the risk of HCC and polymorphisms of the enzymes associated with alcohol metabolism, ADH1B (Alcohol dehydrogenase 1B), and ALDH2 (Aldehyde Dehydrogenase 2) [32]. In addition, alcohol synergizes with other risk factors for HCC, such as diabetes and HCV infection. In patients with excessive alcohol consumption (over 80 g/day), the associated RR of HCC increased from 2.4 to 9.9 among patients with diabetes and increased from 19.1 to 53.9 among patients with HCV infection [33].

### 2.4. Aflatoxin

Aflatoxin is produced by the molds *Aspergillus flavus* and *Aspergillus parasiticus*, which contaminate various types of food such as grains, maize, legumes, and nuts. An estimated 4 billion people globally are at risk of aflatoxin exposure [34]; intake of aflatoxin-contaminated food increases the risk of HCC [35]. A meta-analysis [36] indicated that aflatoxin and HBV infection increased the HCC risk sixfold and elevenfold, respectively, whereas the two risk factors together increased HCC risk 54-fold. The measurement of blood levels of aflatoxin B1–albumin adducts has demonstrated a significant dose-response increase in risks of cirrhosis and HCC among chronic HBV carriers [37]. In addition, serum levels of aflatoxin B1–albumin adducts are an independent predictor of HCC risk in individuals with an acquired HCV infection (odds ratio = 3.65; 95% confidence interval (CI) = 1.32–10.10), even after adjustment for other HCC-related risk factors [38]. Individuals with HBV or HCV infection or considerable alcohol consumption are more susceptible to aflatoxin-associated HCC than those without these risk factors [38].

### 2.5. Obesity, Diabetes, and NAFLD

Excessive body weight is a significant risk factor for several cancer types, and a five-unit increase in body mass index is associated with an RR of liver cancer of 1.59 (95% CI = 1.35–1.87) [39]. In a long-term follow-up study, extreme obesity (body mass index ≥30 kg/m^2^) was observed to be independently associated with a fourfold increased risk of HCC among patients that were anti-HCV-seropositive (RR = 4.13; 95% CI = 1.38–12.4) and with twofold increased risk among patients without HBV or HCV infection (RR = 2.36; 95% CI = 0.91–6.17) after controlling for other metabolic components [40]. In addition, alcohol consumption and obesity exhibit a synergistic association with the risk of HCC development (RR = 7.19; 95% CI = 3.69–14.00). Even after relevant risk factors for HCC are controlled, individuals who consume alcohol and have extreme obesity have a 3.82 times greater risk of developing HCC than those who do not consume alcohol and do not have extreme obesity. The prevalence of diabetes is growing much more rapidly in developing countries than in developed countries, with projected increases of 69% and 20% in developing and developed countries, respectively [41]. Furthermore, type two diabetes mellitus has been observed to be independently associated with the incidence of HCC [42].

The prevalence of the metabolic fatty liver disease is rapidly increasing due to people’s sedentary behavior, low levels of physical activity, and high caloric intake. NAFLD places a substantial burden on public health, affecting 25% of the adult population globally and incurring high healthcare costs [43]. Although the incidence of NAFLD-related HCC is considerably lower than that of other HCC-related etiologies [44], such as chronic HBV and HCV infection, the prevalence of NAFLD is higher than that of chronic viral hepatitis. Therefore, identifying patients with a high risk of developing NAFLD through HCC clinical surveillance is necessary [45].

### 2.6. Family History of HCC

Family history of HCC provides evidence of a direct link between heritable factors and the risk of HCC (Table 1) [46,47,48,49,50,51,52,53,54,55]. However, familial aggregation may be influenced by shared environmental factors and genes. One study examined the family environment risk by estimating HCC risk among spouses [56]. However, no spouse–case correlation was observed for liver cancer, suggesting that the environmental risk factors shared between spouses were limited and that the study may have detected genetic effects contributing to HCC risk. A case-control study revealed that a family history of liver cancer increased HCC risk independently of HBV or HCV infection [55]. A familial predisposition to HCC was investigated among individuals seropositive for HBsAg [54], indicating that HBV carriers with a family history of HCC had a 2.41-fold greater risk of developing HCC (95% CI = 1.47–3.95) compared with those without a family history of HCC. In addition, first-degree relatives of the case group individuals were more likely to have HCC than first-degree relatives of the control group individuals, further suggesting the genetic tendency of HCC. Notably, the synergistic effects of HBV and a family history of HCC multiplied the risk of developing HCC [57]. The synergy between these factors remained significant after stratification by HBeAg serostatus and HBV DNA level. In addition to the positive associations between a family history of HCC and HCC risk, the presence of such a family history is associated with an earlier diagnosis of HCC [58], suggesting that proper clinical monitoring of individuals with a family history is necessary.

### 2.7. Genetic Susceptibility

Genetic susceptibility to HCC is characterized by genetic heterogeneity. Several unlinked genetic defects that are rare in the general population—such as mutations in the genes for a homeostatic iron regulator, alpha 1-antitrypsin deficiency, glycogen storage diseases, porphyria (hydroxymethylbilane synthase and uroporphyrinogen decarboxylase), tyrosinemia (fumarylacetoacetate hydrolase), and Wilson disease (ATPase copper-transporting beta)—increase susceptibility to HCC [59].

### 2.8. GWASs for Identifying Genetic Susceptibility to HCC

The GWAS is a practical and widely used method for discovering genetic predispositions to diseases or other phenotypes [60]. A GWAS maps the patterns of inheritance for the most common form of genomic variation, the single nucleotide polymorphism (SNP). An estimated 10 million common SNPs with a minor allele frequency of at least 5% are transmitted across generations in blocks, which enables a few particular or tag SNPs to capture the vast majority of SNP variation within each block. Several hundred thousand to more than a million SNPs are assayed in a GWAS of thousands of individuals; thus, GWASs are powerful tools for investigating the genetic architecture of complex diseases. In contrast to the candidate gene approach, which explores the association of prespecified genetic variants to specific phenotypes, GWASs test common variants across the human genome to determine associations with disease susceptibility. Therefore, GWASs can be used to discover novel variants and provide valuable insights into the influences of genes on diseases. However, GWASs require testing an enormous number of associations between SNPs and the phenotypes of interest; the thresholds of statistical significance are therefore stringent [61], and a large sample is necessary. In addition, SNPs identified through GWASs generally tend to have an effect of a relatively small to a modest size. The tiered design is an approach frequently used to manage sample sizes. A subset of SNPs determined to be significant in the genome-wide phase (the discovery set) is genotyped in a second tier (the replication set), producing a smaller subset of significantly associated SNPs that are then tested in a third tier (a second replication set), and so forth. We reviewed and summarize several GWASs that were conducted to identify genetic variants associated with HCC, with the summary presented separately for the different chronic hepatitis virus infections.

## 3. GWASs for Identifying Variants Susceptible to HCC in Individuals with Different Viral Hepatitis Status

### 3.1. GWASs for Identifying Variants Associated with HBV-Related HCC

Several GWASs that have examined genetic variants associated with HCC among patients with HBV infection are summarized in Table 2. However, some studies lacked a replication phase following the identification of some variants in the discovery phase [62,63,64]. Zhang et al. [65] conducted a GWAS to investigate genetic susceptibility to HBV-related HCC and identified rs17401966 in the *KIF1B* (kinesin family member 1B) gene, which exhibited a significant association in the discovery phase as well as in 3 replication phases. The protein and mRNA expression of the *KIF1B* gene were significantly elevated in nontumor tissues in rs17401966 G allele carriers, which is consistent with the idea that the *KIF1Bβ* gene can act as a tumor suppressor [66]. One of the replication phases included testing in family trios, in which the significant association of rs17401966 with HBV-related HCC was calculated through transmission–disequilibrium tests. Another GWAS [67,68] involved family trios and revealed rs10272859 in the *CDK14* (cyclin-dependent kinase 14) gene to be associated with HCC among patients with HBV infection. The patients with HBV seropositive HCC carrying the risk G allele of rs10272859 were determined to have a poor prognosis, suggesting that this variant may assist in the improvement of risk stratification and decision-making during early treatment. In regards to familial HCC among HBV carriers, Lin et al. [62] compared patients with HBV-related HCC who had first-degree relatives with HCC to controls with chronic HBV infection and identified 51 SNPs clustered in the *GLUL* (glutamate-ammonia ligase) and *TEDDM1* (transmembrane epididymal protein one) genes that were significantly associated with familial HBV-related HCC.

Several loci in the human leukocyte antigen (HLA) located on chromosome six were determined to be associated with HCC. Furthermore, two SNPs, rs9272105 in the HLA-*DQA1/DRB1* locus and rs455804 in the *GRIK1* (glutamate ionotropic receptor kainate type subunit one) gene, were revealed to be significantly associated with HBV-related HCC [69]. By comparing individuals with HBV natural clearance, defined on the basis of those who were seronegative for HBsAg and anti-HCV antibodies and seropositive for both antibodies to hepatitis B surface antigen and hepatitis B core antibody, with asymptomatic HBV carriers, investigators determined that, in addition to susceptibility to HBV-related HCC, rs9272105 had a negative association with HBV chronic infection. After imputation, the study also revealed that *HLA-DRB1*04:04* and *HLA-DRB1*09:01* were relevant for HCC risk. Another GWAS [70] discovered loci at the *STAT4* (signal transducer and activator of transcription 4) gene and *HLA-DQ*. Similarly, another study [63] identified two SNPs, rs115126566 in the *HLA-DPA1/DPB1* locus and rs114401688 in the *HLA-DQB1/DRB1* locus, associated with the risk of HBV-related HCC; rs114401688 was detected in linkage disequilibrium with rs9272105 in a different study [70]. The *HLA* complex is the most crucial region for humans with respect to infection and innate and adaptive immunity. Imputation methods [71] for predicting HLA genotypes on the basis of SNPs may overcome *HLA* direct genotyping problems, specifically its labor-intensive, time-consuming, and expensive nature. Therefore, imputation approaches have typically been employed when the SNPs near the HLA region are identified through a GWAS [69,70,72].

Chan et al. [68] identified variants in chromosome 8p12 that were associated with HCC among HBV carriers; this chromosome harbors an expressed sequence tag. To control for potential confounding factors, including gender and ethnicity, one study [73] used a sample of solely Chinese men and identified nine SNPs significantly associated with the risk of HBV-related HCC; the two SNPs for which the statistical evidence of an association signal was strongest were rs2120243 in the *VEPH1* (ventricular zone expressed PH domain containing one) gene and rs1350171 in the *FZD4* (frizzled class receptor four) gene. One of the genome-wide significant SNPs, rs4561519 in the *KIF2B* (kinesin family member 2B) gene, is within the same kinesin family as rs17401966 in the *KIF1B* gene, which is consistent with the findings of a related study [65].

In contrast to previously mentioned studies that examined the risk of HBV-related HCC, Zeng et al. [64] recruited participants who were not only infected with HBV but also had elevated serum liver enzymes. The researchers discovered that rs2833856 in the *EVA1C* (EVA-1 homolog C) gene was associated with chronic HBV susceptibility, whereas rs4661093 in the *ETV3* (ETS variant transcription factor three) gene indicated a risk of progression from decompensated cirrhosis to HCC.

**Table 2 viruses-15-00559-t002:** GWASs of HBV-Related HCC Susceptibility.

Study (Authors, Year)	Gene	SNP	OR (95% CI)	Number of Subjects				Comments
				Discovery Phase		Replication Phase(s)		
				Cases	Controls	Cases	Controls	
[65]	*KIF1B*	rs17401966	1.64 (1.49–1.82)	355	360	1962	1430	Includes 159 family trios; HBsAg(+), anti-HBc(+), but anti-HCV(−); Free of HDV and HIV infection; Free of other liver diseases, including autoimmune or toxic hepatitis, primary biliary cirrhosis, and Budd–Chiari syndrome; Replication phase: 4 cohorts with pooled results; rs17401966: associated with the *KIF1B* protein and mRNA expression.
[68]	Non-coding RNA on chromosome 8p12	rs12682266 rs7821974 rs2275959 rs1573266	1.38 (N/A), *p* = 3.76 × 10^−5^ 1.33 (N/A), *p* = 2.32 × 10^−4^ 1.31 (N/A), *p* = 5.19 × 10^−4^ 1.39 (N/A), *p* = 2.71 × 10^−5^	95	97	500	728	All patients are free of antiviral treatment; HBsAg(+) and anti-HBc(+). One of the patients with HCC is anti-HCV(+).
[69]	*HLA-DQA1/DRB1*	rs9272105	1.28 (1.22–1.35)	1538	1465	4431	4725	HBsAg(+), anti-HBc(+), but anti-HCV(-).
	*GRIK1*	rs455804	1.19 (1.12–1.25)					
[70]	*HLA-DQB1/DQA2* *STAT4*	rs9275319 rs7574865	1.49 (1.36–1.63) 1.21 (1.14–1.28)	1161	1353	4319	4966	HBsAg(+), anti-HBc(+), but anti-HCV(−); Free of HDV and HIV infection; Free of other liver diseases, including autoimmune or toxic hepatitis and primary biliary cirrhosis; Replication phase: 6 cohorts with pooled results; Significance determined using p < 5 × 10^−3^ and LD r^2^ < 0.2 between markers.
[73]	*VEPH1*	rs2120243	1.76 (1.39–2.22)	50	50	282	278	Restricted to men; HBsAg(+), anti-HBc(+), but anti-HCV(−); rs4561519 in *KIF2B*: within the same kinesin family as the SNP (rs17401966 in *KIF1B*), as extracted from Zhang [65].
	*FZD4*	rs1350171	1.66 (1.33–2.07)					
	*FZD4*	rs1048338	1.64 (1.31–2.04)					
	*L3MBTL4*	rs2212522	1.57 (1.25–1.97)					
	*FZD4*	rs7116140	1.51 (1.22–1.88)					
	*PCDH9*	rs4480667	1.52 (1.21–1.90)					
	*PRMT6*	rs4417097	1.48 (1.19–1.85)					
	*LHX1*	rs9893681	1.65 (1.22–2.21)					
	*KIF2B*	rs4561519	1.52 (1.14–2.02)					
[62]	*GLUL, TEDDM1*	51 SNPs within 2 SNP clusters, 3 LD blocks	Ranging from 0.14 (0.05–0.41) to 9.87 (3.46–28.16)	139	139	101	N/A	HBsAg(+). Patients with anti-HCV(−) not excluded; Cases: restricted to familial HBV-related HCC; defined as follows: ○ Phase 1 of GWAS: with ≥1 first-degree relatives with HCC; ○ Phase 2 of GWAS: with ≥2 first-degree relatives with HCC; ○ Phase 1 of GWAS: restricted to men; ○ Phase 2 of GWAS: not restricted to individuals with chronic HBV; number of controls not provided.
[67]	*CDK14*	rs10272859	1.28 (1.18–1.38)	537	737	3796	2544	Includes 189 family trios; HBsAg(+), anti-HBc(+), but anti-HCV(−); Free of HDV and HIV infection; Free of other liver diseases, including autoimmune or toxic hepatitis, primary biliary cirrhosis, and Budd–Chiari syndrome; rs10272859: associated with the overall survival of 192 patients with HCC.
[63]	*HLA-DPA1/DPB1* *HLA-DQB1/DRB1*	rs115126566 rs114401688	1.29 (1.19–1.47) 1.64 (1.33–1.96)	1161	1353	N/A	N/A	HBsAg(+), anti-HBc(+), but anti-HCV(−); rs114401688: in LD with an SNP (rs9272105), as extracted from Jiang [70].
[64]	*EVA1C*	rs2833856	2.84 (N/A),p = 1.62 × 10^–6^	214	93	N/A	N/A	HBsAg(+), anti-HBc(+), but anti-HCV(−); rs2833856: associated with risk of HCC among patients with chronic HBV infection; rs4661093: associated with risk of HCC among patients with HBV-related decompensated cirrhosis.
	*ETV3*	rs4661093	2.84 (N/A),p = 2.26 × 10^−6^	214	188	N/A	N/A	

Abbreviations: OR, odds ratio. CI, confidence interval. GWAS, genome-wide association study. HCC, hepatocellular carcinoma. HBV, hepatitis B virus. LD, linkage disequilibrium. N/A, not applicable. SNP, single nucleotide polymorphism. (+), seropositive for. (−), seronegative for.

### 3.2. GWASs for Identifying Variants Associated with HCV-Related HCC

The GWAS approach has been highly successful in determining genetic variants associated with interferon-based antiviral treatment responses [74,75,76] and adverse effects [77,78] during treatment for chronic HCV infection. Discovering genetic variants related to HCC susceptibility may facilitate efficient and targeted surveillance for high-risk patients or prioritizes patients requiring treatment with novel antiviral agents.

In Table 3, we summarize several GWASs that investigated potential genetic variants associated with HCV-related clinical outcomes. Most of these studies examined HCC susceptibility regardless of treatment status. One study [79] compared an HCV-related HCC group with an anti-HCV seronegative control group and identified two genome-wide significant variants, namely rs2596542 in the *MICA* (MHC class I polypeptide-related sequence A) gene and rs9275572 in the *HLA-DQA* and *HLA-DQB* loci. The investigators then compared the HCV-related HCC group to a control group with chronic HCV infection and also compared the chronic HCV infection group to an anti-HCV seronegative control group. The results demonstrated that rs2596542 was associated with progression from chronic HCV infection to HCC and not chronic HCV infection susceptibility. However, rs9275572 was associated with both progressions from chronic HCV infection to HCC and chronic HCV infection susceptibility. Moreover, haplotype analysis revealed that the effect of the risk allele of rs2596542 on HCV-related HCC susceptibility was stronger than that of the risk haplotype of rs2596542 and rs9275572, suggesting that rs2596542 is a principal variant in this region.

In addition, the rs2596542 risk AA genotype was determined to have low serum levels of MICA, supporting a possible biological association. Another GWAS [80] was conducted to compare patients with HCC to those without HCC among patients with HCV infection. The study identified rs1012068, located in the *DEPDC5* (DEP domain containing five) gene, as being significantly associated with HCC among those with HCV infection. The significance remained after adjustment for other covariates, including alcohol consumption, diabetes mellitus, and obesity, and was increased after adjustment for age, gender, and platelet count. The mRNA expression level of the *DEPDC5* gene was significantly higher in the paired HCC tumor and adjacent nontumor liver tissues.

With the increased availability of direct-acting antiviral agents for treating HCV, most patients with chronic HCV may experience treatment-induced RNA clearance resulting from effective antivirals. However, patients with a sustained virologic response (SVR) remain at risk of developing HCC [81]. A study indicated that the three-year and five-year HCC incidence rates after SVR is achieved through interferon-based therapy were 0.5% to 2.0% and 2.3% to 8.8%, respectively [82]. Matsuura et al. [83] conducted a GWAS to investigate possible genetic variants associated with HCC development among clinical patients with chronic HCV infection who achieved SVR, analyzing patients with undetectable HCV RNA 24 weeks after cessation of treatment. The variant rs17047200, located within the intron of the *TLL1* (tolloid-like one) gene on chromosome four, reached genome-wide significance, and the patients who carried the AT or TT genotype had an increased risk of HCC; the adjusted rate ratio was 1.78 (1.17–2.70). The study also revealed that the level of *TLL1* mRNA was increased in a rodent model of liver injury and the liver tissues of patients with fibrosis in comparison with controls.

**Table 3 viruses-15-00559-t003:** GWASs that Identified SNPs Associated With Susceptibility to HCV-Related HCC or Liver Cirrhosis.

Study (Authors, Year)	Phenotype	Gene	SNP	OR (95% CI)	Discovery Phase		Replication Phase(s)		Comments
					Cases	Controls	Cases	Controls	
[79]	HCV-related HCC	*MICA*	rs2596542	1.39 (1.27–1.52)	721	3890	673	2596	Cases: anti-HCV(+) but HBsAg(−); Controls: anti-HCV(−). Also free of cancer, diabetes, and tuberculosis.
		*HLA- DQA/DQB*	rs9275572	1.30 (1.19–1.42)					
[80]	HCV-related HCC	*DEPDC5*	rs1012068	1.75 (1.51–2.03)	212	765	710	1625	Restricted to individuals aged over 55 years; Anti-HCV(+) but HBsAg(−); *DEPDC5* mRNA expression; Significantly different in tumor and nontumor tissues; Not associated with rs1012068 in either HCC cases or the controls.
[83]	HCV-related HCC	*TLL1*	rs17047200	2.37 (1.74–3.23)	123	333	130	210	Restricted to SVR after interferon-based treatment for chronic HCV infection; Anti-HCV(+) but HBsAg(−); Also excluded history of HCC, decompensated cirrhosis, autoimmune hepatitis, primary biliary cirrhosis, and HIV infection; Survival analysis based on prospective data: rs17047200 is associated with the cumulative incidence of HCC.
[84]	HCV-related HCC	*HLA-DQB1*	rs9274684 rs9275521 rs2647046 rs6928482 rs2856723 rs9275086 rs9275210 rs2858324	1.54 (1.36–1.74) 1.56 (1.35–1.80) 1.56 (1.35–1.80) 1.95 (1.73–2.19) 2.68 (2.32–3.09) 2.10 (1.84–2.40) 1.36 (1.20–1.54) 2.42 (2.11–2.78)	520	749	669	429	Anti-HCV(+) but HBsAg(−); Survival analysis based on prospective data; HCV genotype serves as an effect modifier for *HLA-DQB1* in relation to HCC risk.
[85]	sALT(U/L) of anti-HCV(+) individuals:	*BUB1B*	rs568800		803		486		Anti-HCV(+) but HBsAg(−); GWAS: baseline quantitative sALT; Replication phase: grouped into 3 levels of serial sALT; Survival analysis based on prospective data: rs568800 is associated with the cumulative incidence of HCC.
	Persistently ≤15			1 (Reference)					
	Ever >15 but never ≤45			1.41 (1.11–1.78)					
	Ever >45			1.86 (1.34–2.60)					
[86]	HCV-related liver cirrhosis	*HLA*	rs910049 rs3135363	1.46 (N/A), p = 9.15 × 10^−11^ 1.37 (N/A) *p* = 1.45 × 10^−10^	682	1045	936	3800	Liver cirrhosis among patients with anti-HCV(+); Anti-HCV(+) but HBsAg(−).

Abbreviations: OR, odds ratio. CI, confidence interval. GWAS, genome-wide association study. HCC, hepatocellular carcinoma. HCV, hepatitis C virus. HIV, human immunodeficiency virus. SNP, single nucleotide polymorphism. SVR, sustained virologic response. sALT, serum alanine transaminase. (+), seropositive for. (−), seronegative for. ‡ All studies listed in this table were conducted on the Japanese population, except those of Lee and Liu, which were conducted on the Taiwanese population.

A Taiwanese study applied the genome-wide approach and explored potential SNPs associated with HCC among patients with HCV infection [84]. All of the patients were seronegative for HBsAg. Initially, 502 patients with HCC were compared with 749 patients without HCC. The investigators identified a cluster of eight SNPs with significantly different distributions between the two groups. These SNPs were replicated in external samples successfully. The SNPs were located close to the HLA gene on chromosome six, which is a highly polymorphic and complex region. The study performed high-resolution *HLA-DQB1* genotyping to determine the associated risk for HCC incidence in one cohort consisting of patients with anti-HCV seropositivity. The variants on *HLA-DQB1*0602* and *HLA-DQB1*0301* were significantly associated with HCC after long-term follow up. The adjusted rate ratios for the association with HCC were 0.45 (0.30–0.68) and 2.11 (1.34–3.34) for *DQB1*03:01* and *DQB1*06:02*, respectively.

One GWAS involved treating the serum alanine transaminase (ALT) level, a seromarker for liver inflammation, as a quantitative trait and used 564,464 SNPs to examine SNP–ALT associations among 803 anti-HCV seropositives [85]. Twelve SNPs were potentially associated with ALT levels upon the first investigation and were further examined to test their associations with serial ALT levels during a follow-up. Among the study participants, the serial ALT levels were categorized into the following three groups: 158 (19.7%) patients with an ALT level persistently ≤15 U/L, 327 (40.7%), patients with an ALT level consistently >15 U/L and never > 45 U/L, and 318 (39.6%) patients with an ALT level consistently >45 U/L during the follow up. In total, 4 of the 12 SNPs were significantly associated with long-term ALT levels, however, only one (rs568800) was successfully validated among patients with HCV infection in an external population. The A allele (vs. the C allele) of rs568800 was associated with an ALT level >15 U/L but ≤45 U/L and an ALT level >45 U/L, with the adjusted odds ratios being 1.41 (1.11–1.78) and 1.86 (1.34–2.60), respectively. Notably, the associations were pronounced among individuals that were anti-HCV seropositive but seronegative for HCV RNA. The study implied that rs568800 must be assessed in clinical patients with treatment-induced RNA clearance. Finally, patients carrying the A allele had increased HCC risk. With the CC genotype as the reference group, the adjusted hazard ratios for HCC were 2.09 (0.90–4.89) for the genotype AC and 2.64 (1.13–6.17) for the genotype AA.

### 3.3. GWASs for Identifying Variants Associated with HCC Unrelated to Chronic Hepatitis Virus Infection

Table 4 summarizes the result of several GWASs investigating HCC-associated SNPs that are not related to viral hepatitis, and these studies were conducted in different populations [86,87,88,89,90]. In one study [90], general controls were compared to nonalcoholic steatohepatitis-derived HCC (NASH-HCC), which was diagnosed through the histological identification of nonalcoholic steatohepatitis features in nontumor liver tissue in patients with HCC. The study identified the statistically significant SNPs: rs2896019 in the *PNPLA3* (patatin-like phospholipase domain containing three) gene and rs17007417 in the *DYSF* (dysferlin) gene. *PNPLA3* has been consistently reported to be a strong genetic determinant for NAFLD, whereas rs17007417 in the *DYSF* gene is a novel locus associated with NASH-HCC. The study then compared the genotype distributions of these two SNPs in patients with NASH-HCC, general controls, and four subgroups of patients with NAFLD, which were subclassified in accordance with a long-term histological progression indicated by steatosis, steatonecrosis, steatohepatitis, and steatohepatitis with fibrosis. The results revealed a strong association of *PNPLA3* with NASH-HCC and steatohepatitis with fibrosis in one of the subgroups compared with the general control groups. However, no associations between *PNPLA3* and less severe types of NAFLD—namely steatosis, steatonecrosis, and steatohepatitis—were discovered compared with general controls. These results indicated that the *PNPLA3* gene might predispose patients in the later stages of NAFLD to HCC.

A large-scale GWAS in a Japanese population identified novel susceptibility loci across different diseases [89]. The investigators compared patients with HCC with general controls and revealed that rs8107030, located between the *IFNL3* (interferon lambda three) and *IFNL4* (interferon lambda four) genes, was associated with HCC susceptibility. Patients with cancers of the esophagus, stomach, colon, breast, or pancreas were excluded from the control group; however, the presence of a substantial number of patients with type two diabetes mellitus (T2DM) and coronary artery disease (CAD) in the control group may have led to erroneous correlations with the risk alleles of T2DM and CAD. The authors claimed that the results were not biased, as excluding all patients with T2DM or CAD from the control group did not affect the effect size estimates.

All of the previously mentioned studies were conducted in Asian populations, whereas Trépo et al. [87] performed a GWAS on a European population to investigate the risk of HCC development in patients with alcohol-related liver disease. They discovered four significant SNPs: the novel variant rs708113 in the *WNT3A-WNT9A* (Wnt family member 3A-Wnt family member 9A) gene and three SNPs located in genes known to be associated with alcohol-related liver disease, cirrhosis, or HCC, namely the *PNPLA3*, *TM6SF2* (transmembrane 6 superfamily member two), and *HSD17B13* (hydroxysteroid 17-beta dehydrogenase 13) genes.

## 4. Limitations of GWAS

There are limitations to GWAS for investigating genetic variants predisposing to certain diseases. First, SNPs were used as genetic markers in GWAS to identify loci at risk of HCC, however, the causal variants for HCC are as yet unknown. Multiple genetic variants due to linkage disequilibrium facilitate the identifications of genotype-phenotype associations, but the associations may not necessarily pinpoint causal variants. Fine mapping will still be helpful in identifying the causal variants with biological relevance.

Second, there may be false-positive associations in GWAS. Thus, adjustments for multiple testing, as well as careful quality control on ancestry and relatedness, are required to distinguish true signals from false positive ones. Moreover, validating the associations of genetic variants and diseases of interest in an external population of adequate sample size or further meta-analyses can help address the problem of false positives. Therefore, this highlights the importance of collaborations between institutes.

Lastly, the effect sizes of the correlations between SNPs and diseases of interest are usually small. There are methods that view the effect of multiple genetic variants as a whole to address the problem of small effect sizes of correlations to diseases. One method is the polygenic risk scores (PRS) [91], which sums up the effect of disease risk alleles carried by an individual, while another method is combining transcriptomics databases with gene set enrichment analysis [92], which identifies classes of genetic variants discovered in GWASs that are over-represented in a specific trait. These approaches may benefit in identifying individuals with an increased risk for the interested disease.

## 5. Applications of GWAS-identified Genetic Variants

The GWAS approach has demonstrated its efficacy in identifying genetic variants that are associated with the risk of HCC among individuals infected with either HBV or HCV. A number of studies have subsequently employed GWAS-identified SNPs for risk stratification in the context of antiviral treatment for viral hepatitis. For instance, a GWAS identified the SNP rs2596542, located within the *MICA* gene, as a risk of HCV-related HCC [79]. A subsequent study found that the risk genotype of rs2596542, in conjunction with serum *MICA* levels, predicts HCC development among cirrhotic HCV-infected patients who did not achieve SVR after interferon-based treatment [93].

In the era of direct antiviral agents (DAAs), there is still an estimated 5% of HCV patients not achieving SVR, and the impact of DAAs on HCC occurrence or recurrence remains conflicting [94]. Several studies have been conducted on HCV-infected patients receiving DAAs, which have employed either a genome-wide association study (GWAS) or a candidate gene approach to identify genetic variants associated with risk factors for the development of HCC, such as non-SVR rate, elevation in serum ALT levels, or changes in liver stiffness [95,96,97].

Early diagnosis and therapeutic interventions for HCC occurrence following antiviral treatments are essential for ameliorating liver-associated mortality. The GWAS-identified genetic variants at risk of viral hepatitis-related HCC, as summarized in this review, can be utilized in future studies incorporating either candidate gene approach or polygenic risk score to improve risk assessment and precision medicine for individuals infected with viral hepatitis who are receiving antiviral therapy.

## 6. Conclusions

The extent to which specific risk factors contribute to the HCC burden can be estimated using population-attributable fractions (PAFs), which depend on the relationship between the risk factors and HCC and their level of influence, as well as the prevalence of that risk factor in the population. The global PAF of HCC has been estimated as 56% for HBV and 20% for HCV [98]. For obesity, the global PAF is estimated to be 9%, although it is notably higher in North America (24%) than in Southeast Asia (4%) or Sub-Saharan Africa (4%) [5]. However, in addition to these major risk factors, genetic variants also play a role in the development of HCC among individuals who are exposed to these risk factors. The genetic profile identified through GWASs could be used to identify biomarkers, which will enhance the screening and management of HCC. This will be achieved by providing individual risk-based personalization and stratification [99]. In addition, an understanding of the genetic mechanisms underlying individuals’ susceptibility to HCC could lead to improvements in HCC prevention and early diagnosis, as well as the development of new treatments. To accurately differentiate patients with HCC from controls, the SNPs identified through GWASs should be further integrated into conventional risk prediction models. Areas under ROC curves that represent the true- and false-positive rates for the presence of HCC, or the net reclassification improvement, are crucial for the assessment of the prediction values of these genetic variants. Another emerging approach is the development of polygenic risk scores, which reflect risk accumulation by considering multiple SNPs [91]. The polygenic risk score is calculated as a weighted sum of the disease risk alleles carried by an individual. However, there is limited evidence regarding the clinical applications of polygenic risk scores in the context of HCC. The translation of genomic discoveries into practical health benefits requires collaboration between multiple biomedical fields, such as genomics, molecular biology, clinical medicine, bioinformatics, and implementation research. This collaboration is necessary in order to effectively turn the discoveries made in genomics into tangible health outcomes for individuals.

## Figures and Tables

**Table 1 viruses-15-00559-t001:** Family History of HCC.

Study (Authors, Year)	Number of Subjects		OR (95% CI)	Comments
	Cases	Controls		
**Case-control study**				
[51]	229	266	1.60 (0.70–4.80)	Adjusted variables are age and sex; Family members with HCC are parents and siblings; No differences between cases and controls in the family history of malignancies other than liver cancer.
[46]	200	200	4.59 (1.02–20.75)	Adjusted variables are age, alcohol consumption, smoking status, and HBV; Family members with HCC are parents, siblings, children, spouses, and spouse’s siblings and parents; The cohort is composed of men only; Familial tendency may result from shared environmental exposure to, for example, dietary mycotoxin or contaminants.
[49]	204	410	3.80 (1.60–8.90)	Adjusted variables are age and sex; Family members with HCC are parents, siblings, and children; Positive family history of liver cancer, especially a familial clustering of HBV infection in mother-to-child transmission, is associated with HCC risk.
[53]	320	1408	2.90 (1.50–5.30)	Adjusted variables are age, sex, area of residence, education level, alcohol consumption, smoking status, and personal cirrhosis or hepatitis; Family members with HCC are parents, siblings, and children.
[47]	284	464	2.30 (1.20–4.50)	Adjusted variables are age, sex, date of participation, area of residence, education level, alcohol consumption, and HBV or HCV infection; Family members with HCC are parents, siblings, and children.
[52]	111	424	3.98 (1.26–12.59)	Adjusted variables are age, sex, alcohol consumption, and smoking status; Family members with HCC are parents, siblings, and children; The synergistic effect for HCC development between the Pro allele of p53 Arg72Pro and family history of HCC in individuals who are HBV-negative (OR = 11.81; 95% CI = 2.59–53.84).
[48]	347	1075	4.10 (1.30–12.90)	Adjusted variables are age, sex, ethnic group, education level, alcohol consumption, smoking status, and diabetes mellitus; Family members with HCC are parents, siblings, and children; HBV or HCV infection and family history of liver cancer indicate a higher risk of HCC (OR = 61.9; 95% CI = 6.60–579.70).
[55]	229	431	2.38 (1.01–5.58)	Adjusted variables are age, sex, education level, alcohol consumption, smoking status, and HBV or HCV infection; Family members with HCC are parents, siblings, and children; Risk estimates in this study are consistent with the pooled RR from the meta-analysis of published data.
**Cohort study**				
[54]	553	4684	2.57 (2.03–3.25)	Adjusted variables are age and sex of relatives; Family members with HCC are parents, siblings, and children; The cohort is composed of men who are HBsAg-positive; The ORs for HCC with history of HCC are separated in accordance with the type of relative.
[50]	173	240	2.58 (1.01–6.60)	Adjusted variables are age, sex, ethnic group, AFP, and cirrhosis; Family members with HCC are parents, siblings, children, grandmothers, great-grandmothers, grandfathers, uncles, nephews, and half-brothers; The cohort comprises individuals who are HBsAg-positive.

Abbreviations: OR, odds ratio. CI, confidence interval. HBsAg, hepatitis B surface antigen. HCC, hepatocellular carcinoma. HBV, hepatitis B virus. HCV, hepatitis C virus. Pro, proline. Arg, arginine. AFP, alpha-fetoprotein. RR, relative risk.

**Table 4 viruses-15-00559-t004:** GWASs of the Susceptibility of HCC Unrelated to Viral Hepatitis.

Study (Authors, Year)	Gene	SNP	OR (95% CI)	Number of Subjects				Comments
				Discovery Phase		Replication Phase(s)		
				Cases	Controls	Cases	Controls	
[87]	*WNT3A-WNT9A*	rs708113	0.72 (0.64–0.80)	775	1332	874	1059	Alcohol-related liver disease: history of alcohol consumption of >60 g/day over 10 years, together with increased AST and ALT or cirrhosis; European cohort.
[88]	*PNPLA3*	rs2281135	1.64 (1.39–1.95)	239	512	466	943	General controls without a history of viral hepatitis or other liver diseases; Significance remains after adjustment for body mass index and hepatitis virus infection; Moderate-to-high heterogeneity among replication cohorts; Multiancestral: the United States, Japan, Singapore, and Taiwan.
		rs2896019	1.60 (1.35–1.89)					
		rs4823173	1.61 (1.36–1.90)					
	*SAMM50*	rs3761472	1.55 (1.30–1.83)					
		rs3827385	1.49 (1.26–1.76)					
[89]	*IFNL3, IFNL4*	rs8107030	1.44 (1.28–1.62)	1866	19574	N/A	N/A	Status of HBV and HCV infection not mentioned; Excluded patients with esophageal, gastric, colorectal, breast, or pancreatic cancer because of potential confounding; Japanese.
[90]	*PNPLA3*	rs2896019	3.37 (2.21–5.14)	58	7672	N/A	N/A	NASH-HCC susceptibility among the general controls; NAFLD diagnosed through liver biopsy; No history of HBV or HCV infection; Japanese.
	*DYSF*	rs17007417	2.74 (1.85–4.06)					

Abbreviations: OR, odds ratio. CI, confidence interval. AST, aspartate aminotransferase. ALT, alanine transaminase. GWAS, genome-wide association study. HCC, hepatocellular carcinoma. NASH, nonalcoholic steatohepatitis. NAFLD, nonalcoholic fatty liver disease. N/A, not applicable. SNP, single nucleotide polymorphism. (+), seropositive for. (−), seronegative for.

## Data Availability

Not applicable.

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
