# Peer review of "Genetic Susceptibility to Hepatocellular Carcinoma in Patients with Chronic Hepatitis Virus Infection"

_viruses, 2023, doi:10.3390/v15020559_

Round 1
Reviewer 1 Report (Previous Reviewer 2)
The authors present a narrative review focusing on the aetiology of HCC, the academic knowledge of GWAS-identified genetic loci associated with HCC and summarize the relevant GWASs in terms of the different chronic hepatitis virus infection statuses.
The title reflects the subject of the manuscript. It presents a quite clear and useful message for clinical practice. It is well written in terms of clarity, style, and use of English language.
A number of small grammatical errors should be corrected throughout the manuscript ie. L111P3 "meta-analysis study" omit the "study" etc.
Also concluding sentence "To translate genomic discoveries into direct health benefits, interaction among several biomedical disciplines—including genomics, molecular biology, clinical medicine, bioinformatics, and implementation research—is required." needs to be revised as it is far too long makes little grammatical sense.
The aetiology section is sufficiently detailed and explains adequately the purpose of this study in the context of published information.
I would propose adding 2.7.1. GWAS subsections as a new section 3.
The conclusion accurately and clearly explains the main result. The length of the manuscript is good.
All references are appropriate and current.
Author Response
Please see the attachment.

Reviewer 2 Report (Previous Reviewer 1)
This is an interesting and important review.
Author Response
Point 1: This is an interesting and important review.
Response 1: Thank you for your comments. We are grateful for your recognition of the significance of this review.
This manuscript is a resubmission of an earlier submission. The following is a list of the peer review reports and author responses from that submission.
Round 1
Reviewer 1 Report
This is a well written and organized review on genetic susceptibility of viral hepatitis-associated hepatocellular carcinoma. This review will be of interest to a large number of readers.
Author Response
Dear editors,
We would like to thank the reviewers for their insightful comments and efforts toward improving our manuscript. We address all of the concerns of the reviews. Please find our point-by-point responses to the reviewers’ comments.
Response to Reviewer 1 Comments
Point 1: This is a well written and organized review on genetic susceptibility of viral hepatitis-associated hepatocellular carcinoma. This review will be of interest to a large number of readers.
Response 1: Thank you for your comments.
Reviewer 2 Report
The authors present a narrative review aiming to summarize the academic knowledge of GWAS-identified genetic loci and their association with occurrence of
HCC. Indeed a number of studies have shown that genome-wide association study is a powerful tool to identify the novel loci for HBV/HCV-induced HCC.
While the subject is interesting I am afraid the authors mainly focus in presenting published outcomes yet not discussing or elaborating on presented outcomes or future perspectives. What do the authors think their work adds to the published literature. The limitations of GWAs are not presented. Additionally how do the published outcomes relate to the world patient population other than Asian populations on which these outcomes primarily refer to. Lastly the work needs to be revised. A discussion section in which the authors should elaborate on the published literature, present limitations and ongoing trials if any is necessary.
Author Response
Dear editors,
We would like to thank the reviewers for their insightful comments and efforts toward improving our manuscript. We address all of the concerns of the reviews. Please find our point-by-point responses to the reviewers’ comments.
Response to Reviewer 2 Comments
Point 1: The authors present a narrative review aiming to summarize the academic knowledge of GWAS-identified genetic loci and their association with the occurrence of HCC. Indeed a number of studies have shown that genome-wide association study is a powerful tool to identify the novel loci for HBV/HCV-induced HCC.
Response 1: Thank you for your comments.
Point 2: While the subject is interesting I am afraid the authors mainly focus on presenting published outcomes yet not discussing or elaborating on presented outcomes or future perspectives. What do the authors think their work adds to the published literature?
Response 2:
Thank you for your comment and suggestion. The genetic profile identified through GWASs could be used to identify biomarkers and thus refine HCC screening or management by enabling individual risk-based personalization and stratification. In addition, understanding the genetic mechanisms underlying individuals’ predisposition to HCC may lead to improvements in the prevention and early diagnosis of HCC and the development of new drugs for treatment. To ensure the ability to differentiate patients with HCC from controls, the SNPs identified in GWASs should be further included in conventional risk prediction models. Areas under ROC curves that represent the true and false-positive rates for the presence of HCC or the net reclassification improvement are crucial for the assessment of the prediction values of these genetic variants. Another emerging approach is the development of polygenic risk scores, which reflect risk accumulation by considering multiple SNPs. These are the points linking the current work to future perspectives. We have described it in the “Conclusion” section.
Point 3: The limitations of GWAs are not presented.
Response 3: Thank you for your comment and suggestion. We have added the suggested content to the manuscript on page 16, lines 59 to 79, under the title “Limitations of GWAS”.
Point 4: Additionally how do the published outcomes relate to the world patient population other than Asian populations to which these outcomes primarily refer?
Response 4: Thank you for your comment and suggestion. Since HCC, especially HBV and HCV-related HCC, is a great disease burden in Asian countries, most of the GWASs identifying genetic variants at risk of HCC were performed on Asian populations. As the reviewer suggested, to generalize the results to worldwide populations, further GWASs on populations other than Asian are needed for validation. Thanks for your comments.
Point 5: A discussion section in which the authors should elaborate on the published literature, present limitations, and ongoing trials if any are necessary.
Response 5: As suggested by the reviewer, we have added a section discussing the limitations of GWASs to the manuscript on page 16, lines 59 to 79, under the title “Limitations of GWAS”. We have described the ongoing trials in the “Conclusion” section, which includes the clinical implication of utilizing the genetic variants in risk of HCC as a biomarker for disease prediction model and risk stratification, as well as shedding light on the possible pathogenesis of HCC that could be validated by functional analyses and laboratory experiments.
Round 2
Reviewer 2 Report
The authors partly addressed my remarks in their revised version. Unfortunately they failed to present an illustrative discussion of the cited studies and how as a whole GWAs actually changes HCC magament in 2022.